# Towards Ready-to-Use Iron-Crosslinked Alginate Beads as Mesenchymal Stem Cell Carriers

**DOI:** 10.3390/bioengineering10020163

**Published:** 2023-01-26

**Authors:** Timothée Baudequin, Hazel Wee, Zhanfeng Cui, Hua Ye

**Affiliations:** 1Department of Engineering Science, Institute of Biomedical Engineering, University of Oxford, Oxford OX3 7DQ, UK; 2Biomechanics and Bioengineering, CNRS, Centre de Recherche Royallieu, Université de Technologie de Compiègne, CS 60 319, 60203 Compiègne, France

**Keywords:** alginate, iron, calcium, beads, cell carrier, stem cells

## Abstract

Micro-carriers, thanks to high surface/volume ratio, are widely studied as mesenchymal stem cell (MSCs) in vitro substrate for proliferation at clinical rate. In particular, Ca-alginate-based biomaterials (sodium alginate crosslinked with CaCl_2_) are commonly investigated. However, Ca-alginate shows low bioactivity and requires functionalization, increasing labor work and costs. In contrast, films of sodium alginate crosslinked with iron chloride (Fe-alginate) have shown good bioactivity with fibroblasts, but MSCs studies are lacking. We propose a first proof-of-concept study of Fe-alginate beads supporting MSCs proliferation without functionalization. Macro- and micro-carriers were prepared (extrusion and electrospray) and we report for the first time Fe-alginate electrospraying optimization. FTIR spectra, stability with various mannuronic acids/guluronic acids (M/G) ratios and size distribution were analyzed before performing cell culture. After confirming literature results on films with human MSCs, we showed that Macro-Fe-alginate beads offered a better environment for MSCs adhesion than Ca-alginate. We concluded that Fe-alginate beads showed great potential as ready-to-use carriers.

## 1. Introduction

Cell expansion at the clinical scale is a critical need to offer fast, reliable and cost-effective cell therapies for a large variety of diseases [1,2]. Different cell types such as lymphocyte T cells [3] or mesenchymal stem cells (MSCs) [4] require billions of cells per procedure to ensure significant healing, but only a small fraction can be extracted from the patient in the perspective of autologous therapies. In particular, MSCs being adherent cells, relevant culture substrates should be developed to improve their proliferation rates.

Among various structures and materials, polymer beads have shown increasing interest as in vitro cell culture substrates, in particular due to the extremely high surface/volume ratio they offer compared to flat surfaces [5,6,7,8,9]. It could, therefore, be possible to culture larger numbers of cells while reducing culture medium volumes and lab space, decreasing in turn time and cost. Another advantage is the interconnected macro-porosity of the bead bed when carriers are packed together compared to bulk porous materials hosting closed pores. Moreover, they can be used both as cell carriers in static systems and as fluidized bed in perfused bioreactors to benefit from mechanical signals and automatic nutrients renewal [10,11,12,13]. After cell expansion, cells can either be detached from the bead surface to be injected or used for further tissue-engineering approaches [6], or be implanted with the beads into defects as a hybrid cell/biomaterial composite [14]. We focus, in this paper, on the first of these applications.

Commercial microcarriers are available, such as Cytodex^®^, and a broad range of polymers and composites are reported in the literature for the preparation of homemade beads, such as polycaprolactone [6] or alginate [7]. It is sometimes difficult to compare their efficiency due to important variations in terms of composition, size or application protocols. Among these polymers, calcium alginate (sodium alginate crosslinked with calcium ions) has been widely studied as carriers for in vitro cell expansion. Sodium alginate, a biopolymer derived mainly from brown algae, is indeed cheap, easy to store and prepare and biocompatible [15]. It can be easily crosslinked with calcium ions Ca^2+^ from Calcium Chloride (CaCl_2_) dissolved in water to form calcium alginate (Ca-alginate) beads with high throughput using various fabrication methods [16,17,18]. Unfortunately, although Ca-alginate is biocompatible (no mutagenicity, cancerogenicity or cytotoxicity), it is poorly bioactive: cell adhesion and proliferation on the surface of pure Ca-alginate materials is extremely limited [7,19]. Blends or functionalization with more bioactive substances, such as chitosan [7], collagen [16], cellulose [20], calcium phosphates [21] or fibronectin [22], are therefore needed to trigger a positive cell response. This results in more complex fabrication protocols, with increases in labor work, costs and batch-to-batch variations.

However, sodium alginate can be crosslinked with other ion sources to form scaffolds with alternative biological and physicochemical properties, such as iron chloride (FeCl_3_), approved by the Food and Drug Administration FDA for food industry [23]. In particular, iron alginate (Fe-alginate) films and porous structures were previously proposed and showed promising results regarding cell adhesion and proliferation, but were tested with fibroblasts only and in a very limited number of studies [24,25]. Such materials were reported more frequently for drug release [26,27,28,29] or the treatment of wastewater [30,31], but not for in vitro MSCs culture experiments. In other studies, it was confirmed that Fe-alginate solutions could form macro-sized beads, but biological characterization of the beads was lacking as well [32].

This paper reports therefore the first study, to the best of our knowledge, investigating the development and the biological characterization of Fe-alginate beads and their potential for in vitro mesenchymal stem cell culture. It would be of primary interest to produce “ready-to-use” iron alginate beads, as easy to prepare as calcium alginate beads, but with relevant bioactivity without the need for coating or extra preparation steps. We compared two different sizes (micro- and macro-carriers) of Fe-alginate beads with Ca-alginate ones and 2D culture surfaces as controls.

After confirmation of the potential of Fe-alginate films as reported in literature, we (1) analyzed the effect of the alginate composition (M-to-G chain ratio), (2) optimized for the first time the fabrication of micro-sized Fe-alginate beads through electrospray and (3) assessed the early cell response of MSCs cultured on the bead surface. The hypothesis of this proof-of-concept study was that alginate solution crosslinked with iron chloride would be promising to form ready-to-use stem cell carriers (i.e., without compulsory functionalization steps) that could decrease costs and labor time for cell expansion systems.

## 2. Materials and Methods

### 2.1. Materials

Sodium alginate powders with low G-chain content (Low-G, 39.1%) and high G-chain content (High-G, 65–75%), calcium chloride (CaCl_2_) and iron chloride (FeCl_3_) were purchased from Sigma-Aldrich (USA). Information regarding the M/G ratio were provided by the manufacturer.

### 2.2. Fabrication of Alginate Beads

Alginate solutions at 1% and 2% were prepared by dissolving 100 mg and 200 mg of sodium alginate in 10 mL of 0.9% sodium chloride (Sigma-Aldrich, Burlington, MA, USA) solution, respectively. Iron chloride crosslinker solution at 1% was prepared by dissolving 100 mg in 10 mL of demineralized water. Calcium chloride crosslinker solution at 102 mM was prepared by dissolving 170 mg in 20 mL of demineralized water.

A homemade electrospray set-up (Figure 1) was used for the fabrication of micro- and macro-sized alginate beads (room temperature and humidity not controlled). Briefly, the alginate solution was perfused with a syringe pump at constant flow rate through a needle with a flat tip above a metallic ring and a Petri dish (10 cm diameter) containing the crosslinker bath (either calcium or iron chloride). High voltage was applied between the needle and the metallic ring to trigger electrospraying. Stirring was used during the process (60 RPM, stirring bar around 2 cm long) and beads were then maintained in the crosslinker bath for 30 minutes before washing with PBS (Phosphate Buffer Saline, Gibco, Billings, MT, USA) three times. Beads were then stored in PBS at 4 °C before use.

For the fabrication of sterile beads in order to perform biological analysis, sodium alginate was dissolved in sterile water upon heating at 70 °C for 30 min (pasteurization). Crosslinker solutions were sterilized by filtering them through 0.22-µm syringe filters. The electrospray set-up and all tubing were cleaned with 70-% ethanol and washed thoroughly with sterile demineralized water. All glassware was autoclaved and beads were washed with sterile PBS in a biosafety cabinet (sterile laminar air flow) immediately after crosslinking. All steps were performed in a cell culture platform with restricted access.

Optimal parameters (voltage, flow rate and needle gauge) for the fabrication of micro- and macro-sized beads are reported in Table 1. Parameters for the fabrication of calcium-based beads and Fe-Macro were based on previous studies [7]. Detailed results of the optimization of Fe-Micro fabrication are reported in the Results and Discussion section with alternative parameters (Flow rate: 1.0, 1.5, 1.7 mL/Min; Voltage: 10, 12 kV).

### 2.3. Alginate Films Fabrication

The same solutions (sodium alginate and crosslinkers) were used for the fabrication of alginate films as described by Machida-Sano et al. [24]. The alginate solution was poured in wells of a 24-well plate and dried overnight at 60 °C (2 mL/well). The wells were then covered by calcium or iron chloride solution (1 mL/well) and incubated for 30 min at room temperature. Stable millimetric structures, visible to the naked eyes, were therefore formed in the wells. These films were then washed three times with demineralized water and stored at 4 °C in PBS before use. Prior to cell seeding, films were washed with 70 % ethanol for 30 min then rinsed thoroughly with sterile PBS.

### 2.4. Alginate Bead Characterization

Phase contrast images were acquired using an Eclipse Ti (Nikon, Tokyo, Japan) microscope, and bead diameters were measured with the software NIS-Elements (Nikon, Japan) using the “intensity profile” tool. Briefly, a linear section is drawn on the images through the center of a bead and the corresponding intensity histogram is plotted, allowing for the measurement of the diameter (length between the intensity shifts).

The Fourier transform infrared spectroscopy (FTIR) spectra were acquired from raw sodium alginate powders and from crosslinked beads using an FTIR spectrometer (Bruker, Tensor 27, North Chicago, IL, USA) equipped with attenuated total reflectance (ATR, Pike, Madison, WI, USA).

To assess the stability of the beads over time, Fe-Macro samples prepared with the different sodium alginate references were maintained in 12-well plates in 2 mL of PBS or culture medium (DMEM low glucose, Gibco, USA) for one week at 4 °C or 37 °C. Ca-Macro beads were used as control. Pictures of the macroscopic behavior of the beads were acquired at day 7 with a Fujifilm XQ1 camera (Japan).

### 2.5. Cell Culture and Seeding Procedures

Green Fluorescence Protein (GFP)-producing human MSCs were kindly provided by the Department of Paediatrics and Adolescent Medicine, LKS Faculty of Medicine, The University of Hong Kong. MSCs were precultured in T75 flasks in DMEM culture medium (Gibco, USA), supplemented with 10% fetal bovine serum (Gibco, USA) and 1% penicillin-streptomycin (Gibco, USA) at 37 °C, 5% CO_2_ in humidified atmosphere. Cells were then harvested at P10 using trypsin-EDTA (Gibco, USA) to be seeded on alginate films and beads.

On films, cells were seeded in the wells of the 24-well plate hosting the films (200,000 cells per well). Cells were also seeded in empty wells as plastic surface controls. Medium was refreshed every 2–3 days for one week (analysis at day 7).

To culture MSCs on beads, glass bijou vials were coated with Sigmacote^®^ (Sigma-Aldrich, USA) to prevent cells and micro-sized beads from adhering on the walls. Alginate beads were then pre-incubated in complete medium for 48 h in the vials prior to seeding. On macro-sized beads, culture medium was then removed and MSCs were seeded at an initial density of 1 million cells per vial in 500 µL. After two hours, fresh culture medium was added up to 5 mL per vial. On micro-sized beads, medium was refreshed (5 mL) and the same number of MSCs were injected in the beads suspension. Samples were then gently shaken every 15 min for 2 h to homogenize cell seeding.

### 2.6. Cell Behaviour Analysis

The GFP-producing cells were observed with fluorescence microscopy using an Eclipse Ti (Nikon, Japan) microscope. Images were acquired after one week (culture on films) and over time (days 1, 3 and 7, culture on beads). Phase contrast images were also recorded to visualize beads more clearly.

Metabolic activity of cells alone or with Fe-alginate beads (all sizes) was evaluated with the Alamar Blue assay at days 1, 4 and 7 to evaluate the cytotoxicity of released ions. Briefly, the culture medium was replaced by fresh medium supplemented with 10% of Alamar Blue solution (Invitrogen, Waltham, MA, USA) and samples were incubated for 2 h under normal culture conditions. The background controls used as blank for the different conditions were, respectively, (1) Alamar Blue solution incubated alone (no cells, no beads) or (2) Alamar Blue in contact with beads only (no cells). The Alamar Blue solutions were then removed from the samples and the fluorescence intensity recorded at 560/595 nm with a spectrofluorimeter (SpectraMax i3x, Molecular Devices, San Jose, CA, USA). Samples were put back in incubation with fresh culture medium. A high fluorescence intensity highlights important reduction of the dye by mitochondria and therefore high metabolic activity of the samples.

After one week, cells were harvested using trypsin-EDTA and isolated from the beads with 70-µm cell strainers. Cell count and viability were assessed using Trypan Blue (1:1) and an automated cell counter (Countess, Invitrogen, USA). To evaluate cell recovery after expansion on beads, the same process was applied and cells were then put back in 2D culture in T75 flasks in normal culture conditions (initial seeding density 5000 cells per cm^2^) for one additional week. Cells were then harvested from the flask surface with Trypsin-EDTA to check cell viability again with Trypan Blue. Fresh MSCs that had never been cultured on beads were seeded, simultaneously, in flasks with the same density as controls.

### 2.7. Statistical Significance

Quantitative data were plotted as mean and standard error of 3 independent experiments. Statistical significance was tested using non-parametric analysis of variance (no assumption of Gaussian distribution). Results were considered statistically significant for *p* < 0.05 (*** *p* < 0.001, **** *p* < 0.0001). All results were tested for statistical significance, the absence of specific notation in the figures corresponds to the absence of statistical significance. When the absence of significance is highlighted, it appears as “N.S.” in the figures.

## 3. Results and Discussion

### 3.1. Iron Alginate Films Were Confirmed as More Bioactive Substrates Than Calcium Alginate Films

Results previously reported in the literature for fibroblast culture on Fe-alginate films were first repeated to confirm the potential of scaffolds based on alginate crosslinked with iron ions for enhanced cell bioactivity. Initially, Machida-Sano et al. showed that fibroblasts seeded on Fe-alginate films benefited from an appropriate substrate for cell adhesion and development, in contrast with Ca-alginate [24]. Cell number after 18 days was even similar to samples grown on a plastic surface. Machida-Sano et al. hypothesized that these differences in cell behavior could result from the variations of surface properties of the films based on the crosslinking ions [23,24]. Thanks to changes in wettability, Fe-alginate samples would be able to adsorb increased amounts of proteins such as vitronectin and fibronectin, well-known as crucial biomolecules for cell attachment [23,24]. Vitronectin, in particular, could be crucial for the proper attachment of cells on alginate beads and using iron chloride as a crosslinker increases drastically the adsorption of this protein, leading in turn to higher attachment of human dermal fibroblasts than on Ca-alginate [24].

Culture of MSCs on Fe- and Ca-alginate films allowed us to confirm this trend with another cell source for the first time (Figure 2). Samples of Ca-alginate films showed only isolated and poorly spread cells after one week while an interconnected network of MSCs covered the surface of Fe-alginate films. We noticed, however, a difference in confluence between plastic culture surface (T-flasks) and Fe-alginate samples, cells reaching a confluent state faster on plastic. We were able to confirm anyway the rationale for using Fe-alginate materials to enhance cell response compared to Ca-alginate without the need for functionalization steps.

### 3.2. The M/G Chain Ratio of Alginate Has Important Effects on Bead Stability in Culture Conditions

The study of alginate beads macroscopic behavior, namely swelling and degradation in an aqueous environment, was performed on macro-sized beads crosslinked with either iron or calcium chloride. Beads were immersed in either PBS or DMEM and kept in incubation at either 4 or 37 °C for one week.

The chemical structure of sodium alginate is a combination of two different chains, guluronic acid (G) and mannuronic acid (M) [33]. The alginate molecular structure contains blocks of consecutive G or M monomers (-GGG- or -MMM-) or blocks of alternating monomers (-MGMG-). Commercial sodium alginate powder can vary a lot in terms of balance between G- and M-chain, with the exact composition, more or less accurately characterized. The blocks vary considerably in length and distribution depending on what species and part of the seaweed the alginate is extracted from [34]. However, it has been reported that the M-to-G chains ratio (M/G) could have an effect on the mechanical properties of alginate gels, as well as their stability, porosity and, in turn, the biological response, depending on the crosslinker used [34,35]. It is known that the affinity of alginate towards cations increases as the amount of G-blocks present in the alginate structure increases. Gelation occurs through the cooperative binding of divalent cations and the G-block regions. The binding of trivalent cations compared to divalent cations is enhanced because trivalent cations are able to interact with three carboxyl groups at the same time, forming a 3D bonding structure that is much more compact [36].

Here, initially, the use of alginate with a high M/G ratio (i.e., Low-G) led to a very poor stability of the Fe-alginate beads in certain conditions, in particular DMEM at 37 °C (Figure 3). A very important swelling occurred (bead diameter had more than doubled), resulting in poor cohesion of the beads that were immediately destroyed upon mechanical mixing, preventing us from performing additional measurements. It was consistent with the effect of M/G ratio on the stability of crosslinked alginate structures previously reported [33,37,38], with a high G-chain content promoting long-term stability and mechanical cohesion [34,35]. Indeed, switching to sodium alginate with a low M/G ratio for the preparation of the beads modified drastically their behavior (High-G, Figure 3). The swelling was then limited and showed no difference between the various medium and temperature conditions, confirming the effect of the chain ratio. More importantly, beads remained stable with a spherical shape upon mechanical mixing and transfer to new vials. We concluded therefore that the high G-chain content alginate could be used to prepare Fe-alginate beads able to be handled for cell culture protocols in regular culture conditions (DMEM, 37 °C). It can be mentioned that the behavior of the Ca-alginate beads used as reference was not altered by the variations of parameters. They remained stable (all conditions) with limited noticeable swelling (Low-G in PBS), as expected [39].

FTIR analysis was then conducted to evaluate crosslinking and confirm the difference in the M/G ratio between alginate references. The spectra of raw alginate powders and Fe-alginate beads are reported in Figure 4. Ca-alginate samples from high-G alginate are plotted as a reference.

Overall, the shifts occurring between the spectra of raw sodium alginate [20,40,41] and Fe-alginate beads confirmed crosslinking as the two C-O-O stretching peaks around 1400 and 1600 cm^−1^ shifted to lower and higher values, respectively [30,42]. This behavior appears similar for many ions used to crosslink alginate [30] and was indeed noticed also in Ca-alginate spectra, along with changes in the shape of the O-H stretching large peak [43,44]. Differences between high-G and low-G samples mostly occur in terms of relative peak intensities [45].

Therefore, we showed here the influence of the M/G ratio on bead stability and the successful crosslinking of sodium alginate with iron ions. High G-chain content sodium alginate was then used for the rest of the experiments reported in the present paper. It has to be mentioned that swelling still appeared on beads made from high-G sodium alginate, for both iron- and calcium-crosslinked samples. This did not jeopardize cell attachment as long as the swelling was over before cell seeding [35], which could be achieved within 48 h [39]. We used therefore a 48 h pre-incubation period of the beads in DMEM before cell culture experiments. Further investigations could however be performed in the future to limit even further the swelling of Fe-alginate beads, thanks to a refined medium composition or a more accurate, more optimal M/G ratio.

### 3.3. Homogeneously Distributed Iron Alginate Beads Can Be Obtained through Electrospray

The fabrication of macro-sized beads requires only a stable flow rate of alginate solution dropped into a crosslinker bath [46], while decreasing the diameter to 100–200 µm requires systems involving more parameters such as co-axial air flow, vibrating nozzle or electrospray [16,17]. Such parameters need to be balanced and optimized to assess the potential of the method for the fabrication of beads from a given polymer solution with high homogeneity [17].

In particular, electrospraying requires strict control of a number of process variables. Mostly, these variables are selected based on empirical knowledge for optimal bead formation. Maintaining the sphericity of the alginate beads is important for good mechanical and chemical stability. It was reported that the gel bead strength of non-spherical beads was greatly reduced as compared to spherical beads [47]. In addition, the production of stable, spherical beads with narrow size distribution is desirable as it gives good reliability and reproducibility of results obtained. The fabrication of micro-sized Ca-alginate through electrospray is well-known, including from previous studies in our group [7]. In contrast, to the best of our knowledge, we report here the first evaluation of the parameters allowing for the optimal fabrication of the electrosprayed Fe-alginate beads. The optimization of flow rate and voltage had to be performed to ensure relevant bead diameter, size distribution and repeatability.

Results of bead diameter mean and distribution for various flow rates (1, 1.5 and 1.7 mL/min) and voltages (10, 12 kV) are reported in Figure 5A–D. We noticed that the bead diameter decreased with a flow rate from 307 ± 154 µm to 206 ± 101 µm (Figure 5B, *p* < 0.001). More importantly, with a low flow rate, a non-homogeneous, two-peak distribution appeared (Figure 5A), while monodisperse distribution was achieved with 1.7 mL/min. This flow rate was, therefore, selected for the optimization of voltage (Figure 5C,D). The diameter was decreased further by increasing voltage (*p* < 0.0001), and size distribution was improved with a smaller deviation for a higher voltage. These variations in diameter were consistent with results reported in the literature for calcium alginate particles [7,48].

The batch-to-batch reliability was then checked by comparing the size distribution of independent productions of beads, as seen in Figure 5E. This confirmed for the first time the feasibility and repeatability of the fabrication of micro-sized Fe-alginate beads with electrospray.

The complete size distribution analysis among the different groups (iron- or calcium-based alginate, macro- or micro-sized beads) is reported in Figure 6. For a given crosslinker, the difference in mean diameter between macro- and micro-sized beads was successfully confirmed as significant, allowing us to investigate size effects on the biological response. The optimal electrospray parameters for Ca-Micro samples led to a slightly higher diameter (368 ± 64 µm) than Fe-Micro (147 ± 69 µm) samples, but no statistical significance was found between Ca-Macro (2753 ± 294 µm) and Fe-Macro (3167 ± 247 µm) groups. The effect of the crosslinker on cell development could, therefore, be analyzed as well.

### 3.4. Iron Alginate Carriers Enhanced Cell Attachment and Proliferation Compared to Calcium Alginate

We first investigated the possible cytotoxic effects of Fe-alginate beads present in the culture medium on MSCs attached and grown on tissue culture plastic surface (6-well plate), to analyze if the release of specific ions in the environment could alter cell proliferation and viability over time. MSCs were cultured for one week with or without beads dropped in the culture medium. Metabolic activity was evaluated with Alamar Blue over time, then cells were harvested to measure cell number and cell viability at day 7. Results are reported in Figure 7.

Both groups, with or without beads, showed an increased metabolic activity over time, highlighting cell expansion (Figure 7A). Assuming that all cells from the population express on average the same activity, results were indeed directly linked to the amount of living MSCs in the different conditions. Although an initial ion release could have decreased cell activity at day 1, in particular for one independent sample, the cell population in the presence of beads started then to grow normally and no significant difference was found between on at day 7 regarding either metabolic activity or cell viability (Figure 7B). Non-statistically, the metabolic activity of cells cultured in presence of Fe-alginate beads was however found to be quantitatively lower. It was not possible to exclude the lowest sample due to external causes (such as contamination), so we concluded the absence of a significant cytotoxic effect from the Fe-alginate beads on the close environment of cells but could not show a strong equivalence between both groups in this proof-of-concept study.

MSCs were therefore cultured directly on the surface of the beads for one week on Fe-alginate and Ca-alginate of both sizes. Phase contrast (Figure 8A) and fluorescence images (Figure 8B) were acquired at day 1, 3 and 7 to assess initial attachment and proliferation of the MSCs on the bead surface. Overall, as hypothesized, cells were able to expand better on Fe-alginate samples. Although a higher initial cell attachment could be noticed on Ca-Micro on day 1, cells were not able to proliferate further on Ca-alginate samples and barely any cells were visible after one week on Ca-Micro and Ca-Macro. We hypothesized that the cells noticeable on the Ca-micro samples at day 1 were actually poorly attached (round and isolated cells) and spread at the interface between several beads in contact; over time, they were easily detached and killed due to the movements of beads in culture. In contrast, MSCs started to grow on the surface of Fe-alginate beads towards a continuous tissue (Fe-Macro) or in a lesser extent towards dense cell clusters (Fe-Micro). Surprisingly, the higher surface/volume ratio provided by Fe-Micro didn’t lead to a drastic increase in cell proliferation compared to Fe-Macro at day 7. The same behavior was noticed between Ca-Micro and Ca-Macro. As the surface/volume ratio is well-known to enhance cell development [5], our results could highlight cross-effects between this ratio and the carrier composition. There was, here, a stronger influence of the crosslinker that would confirm the higher bioactivity of iron alginate scaffolds, and in particular optimal conditions with macro-sized beads.

From the perspective of cell therapy or tissue engineering, cell expansion on carriers is just a single step in the process and the cells of interest have to be later harvested to be used for a specific application. Therefore, we analyzed the recovery potential of MSCs after growth and harvest from Fe-alginate beads. We were not able to harvest enough cells from Ca-alginate beads (macro or micro) to perform such a study on these groups. After one week, MSCs were detached from the surface of Fe-Micro and Fe-Macro samples with trypsin-EDTA and seeded in regular cell culture flasks. After one additional week of culture in a flask, cell viability was measured and compared to fresh MSCs that had been maintained in 2D culture as control (Figure 9). All cells were found to be at a confluence state or very close (minimum 95%) after one week. No significant differences in viability were found between groups, including between Fe-Micro and Fe-Macro, although higher variability was noticed with Fe-Micro samples and the highest viability was obtained for cells initially harvested from the plastic surface. Results from Fe-Macro and plastic surface were overall very close (both around 90%). This strengthened the choice of Fe-Macro samples as the optimal parameters for cheap, easily produced and ready-to-use carriers for cell expansion. Even if the stem cell viability (before or after recovery) was not statistically different from 2D culture, the surface-to-volume ratio of cell culture on microcarriers remains a major advantage over regular culture protocols to amplify cells at the clinical scale.

## 4. Conclusions

We reported here for the first time the investigation of iron-crosslinked alginate beads as carriers for in vitro mesenchymal stem cell expansion. After optimization and characterization of the biomaterial, in particular stability through analysis of the M/G alginate chains ratio, we showed the higher bioactivity potential of Fe-alginate carriers compared to the well-known Ca-alginate scaffolds, leading to an increase in MSCs attachment and growth. These results validated our hypothesis that alginate crosslinked with iron chloride to form beads is of primary interest towards the development of cheap and ready-to-use cell carriers, compared to Ca-alginate products requiring additional functionalization steps to ensure relevant cell behavior. This study allowed us to show for the first time the production of Fe-alginate beads through electrospray, and to propose macro-sized beads as optimal parameters. Following this proof-of-concept approach, further studies could now investigate long-term use of this biomaterial, regarding cell behavior as well as degradation rate and evolution of mechanical properties. In particular, stemness markers could be investigated to ensure that MSCs remain undifferentiated even after a long culture period. The potential with other cell sources, in particular primary human MSCs, should now be considered, as well as the direct use in bioreactor systems. The outlooks also include the encapsulation of factors and drugs within the beads, to control the in vitro cell development but also for implantable drug delivery systems comprising both cells and beads for the filling of defects.

## Figures and Tables

**Figure 1 bioengineering-10-00163-f001:**
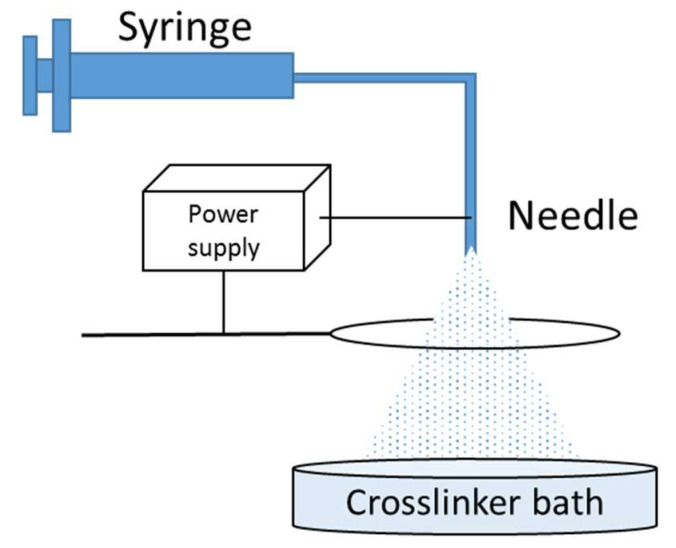
Electrospray set-up.

**Figure 2 bioengineering-10-00163-f002:**
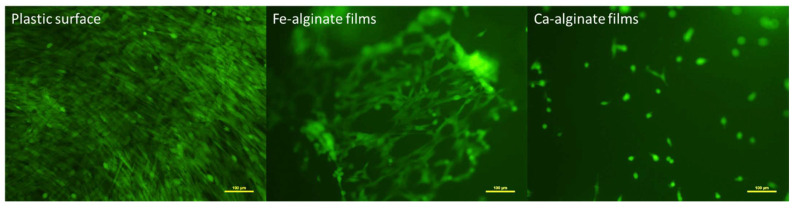
Representative observations of 3 independent cultures of MSCs (fluorescence microscopy) attached on control plastic surface, Fe-alginate or Ca-alginate films (Day 7 after seeding). Cells appear green due to self-produced GFP.

**Figure 3 bioengineering-10-00163-f003:**
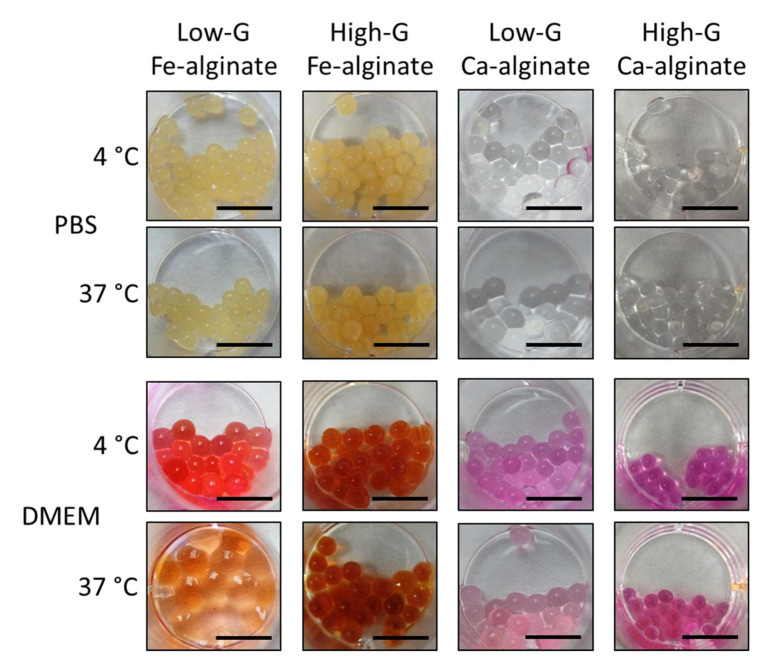
Swelling and stability of Fe-alginate and Ca-alginate beads with different M/G ratios in PBS or DMEM culture medium after one week of incubation at 4 °C or 37 °C. At day 0, all beads showed size and aspect similar to High-G Ca-alginate beads incubated in DMEM at 37 °C. Beads incubated in DMEM appeared red because of internalization of the phenol red present in DMEM. Scale bar: 10 mm.

**Figure 4 bioengineering-10-00163-f004:**
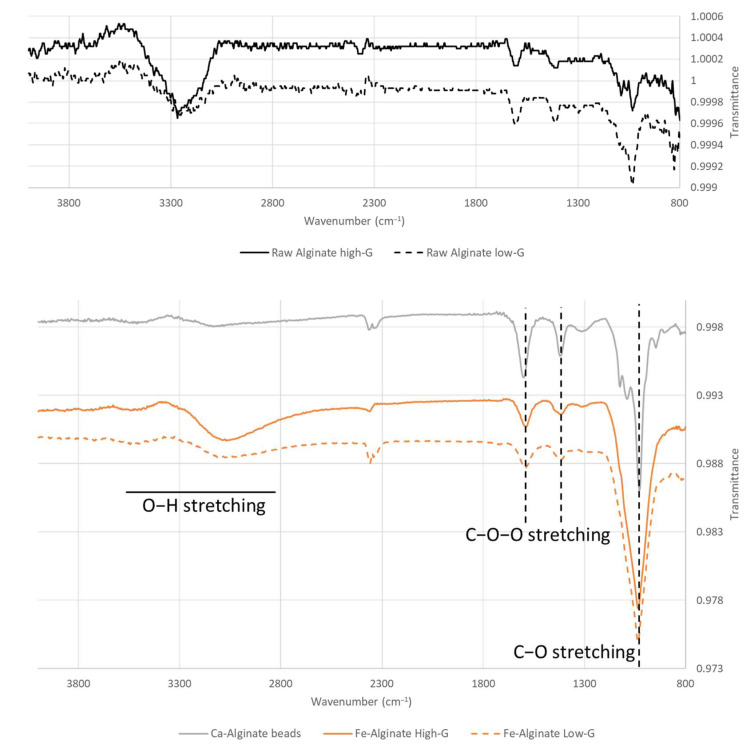
FTIR spectra of raw alginate powders, Ca-Alginate and Fe-Alginate beads for low and high G-content sodium alginate references.

**Figure 5 bioengineering-10-00163-f005:**
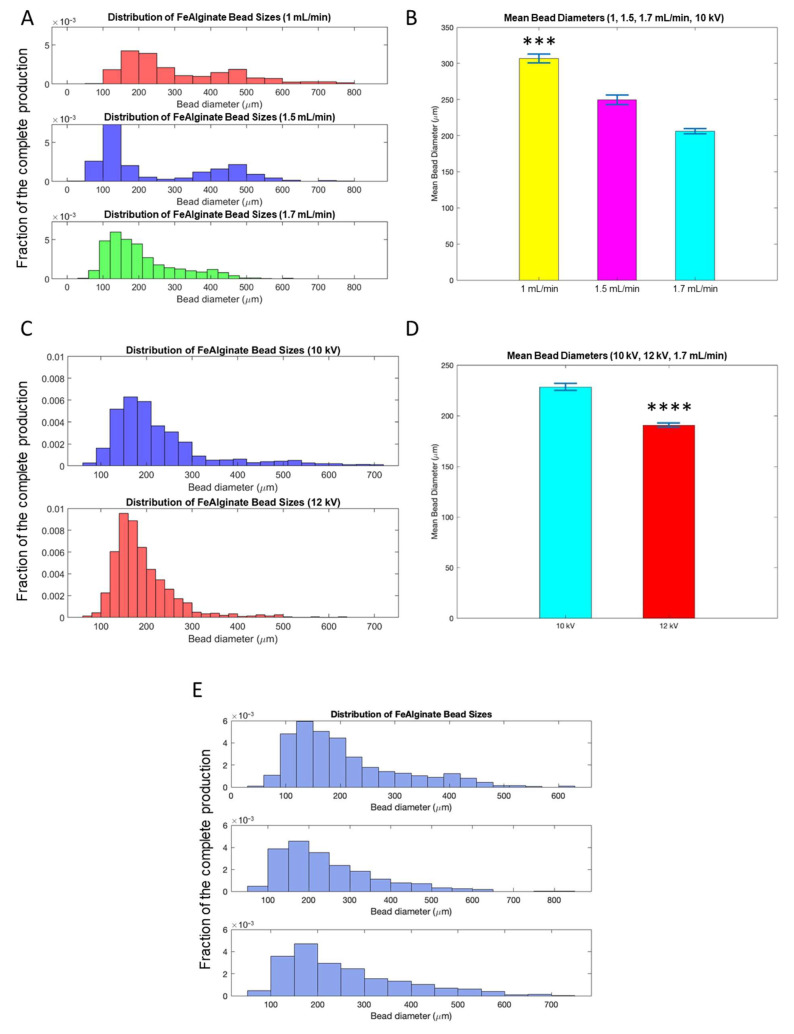
Optimization of electrospray parameters for the production of Fe-Micro beads. Comparison between different flow rates, distribution of bead diameter (**A**) and mean bead diameter in a batch (**B**). Comparison between different voltage, distribution of bead diameter (**C**) and mean bead diameter in a batch (**D**). Distribution of bead diameter of three independent batches (**E**). Mean and standard error of mean. *** *p* < 0.001, **** *p* < 0.0001.

**Figure 6 bioengineering-10-00163-f006:**
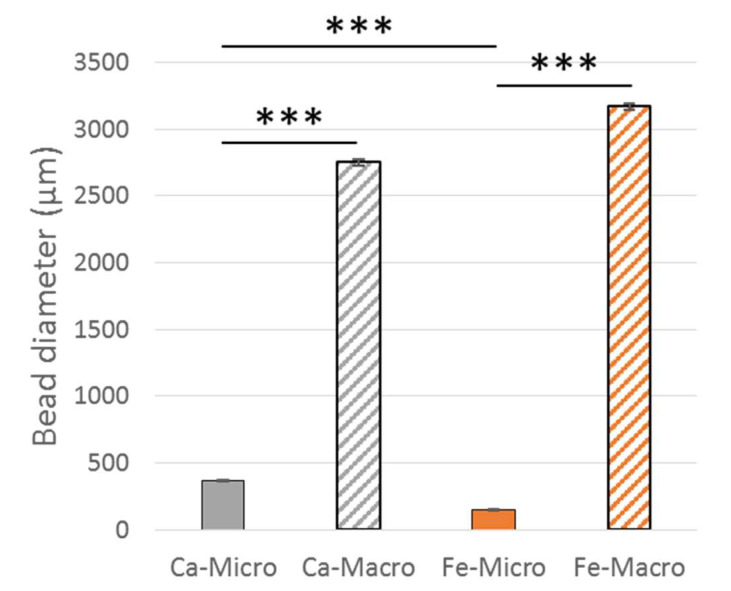
Bead diameter of the different groups. Mean and standard error of mean. *** *p* < 0.001.

**Figure 7 bioengineering-10-00163-f007:**
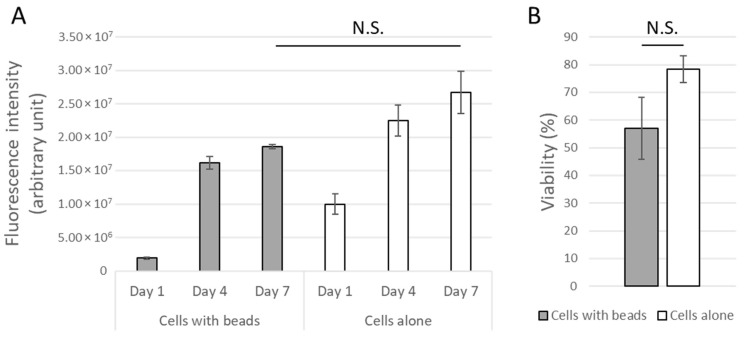
Metabolic activity (**A**) and cell viability (**B**) of MSCs cultured on tissue culture plastic surface with or without Fe-alginate beads dropped in the culture environment. Mean and standard error of mean. No statistical significance was found (N.S.: Non-significant (*p* > 0.05)).

**Figure 8 bioengineering-10-00163-f008:**
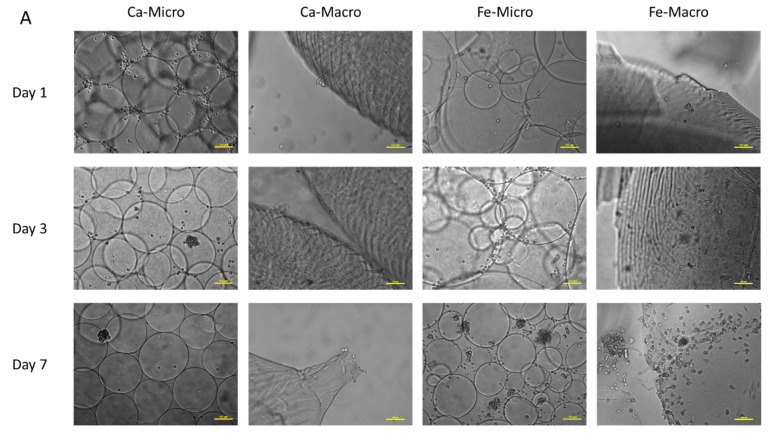
Representative observations of (**A**) phase contrast microscopy of MSCs cultured on alginate beads over time, and (**B**) fluorescence microscopy of MSCs cultured on alginate beads over time. Cells appear green thanks to self-produced GFP. Scale bar: 100 µm.

**Figure 9 bioengineering-10-00163-f009:**
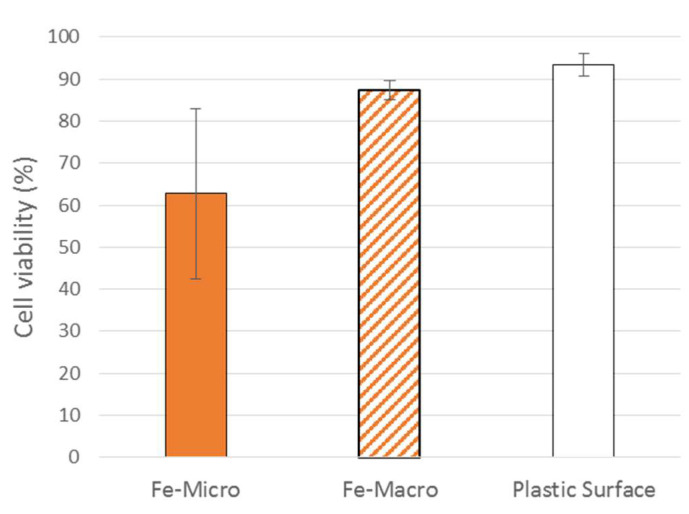
Cell viability of recovering cells cultured in flasks for 7 days after being harvested from Fe-Micro beads, Fe-Macro beads or plastic surface. Mean and standard error of mean. No significant difference between groups (*p* > 0.05).

**Table 1 bioengineering-10-00163-t001:** Optimal electrospray parameters for the production of micro- and macro-sized alginate beads crosslinked with Calcium or Iron Chloride.

Crosslinker	Abbreviation	Voltage	Flow Rate	Needle Gauge
**CALCIUM CHLORIDE**	Ca-Micro	7.5 kV	3 mL/min	30G
	Ca-Macro	/	3 mL/min	19G
**IRON** **CHLORIDE**	Fe-Micro	12 kV	1.7 mL/min	30G
	Fe-Macro	/	3 mL/min	19G

## Data Availability

The data presented in this study are available on reasonable request from the corresponding author.

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
