# Peer review of "Towards Ready-to-Use Iron-Crosslinked Alginate Beads as Mesenchymal Stem Cell Carriers"

_bioengineering, 2023, doi:10.3390/bioengineering10020163_

Round 1

Reviewer 1 Report (New Reviewer)

The paper is very well written and clear. The authors carefully explained the goal of each experiment. The results are significant. The method of homogeneous Fe-alginate microbeads synthesis was developed. Their advantages compared to Ca-alginate beads were demonstrated. I think nevertheless that some aspects of the paper should be improved. First of all, quantitative analysis of results would be valuable in some experiments. Please, find my comments below.

1.           Line 18 – what is M/G ratio?

2.           FeCl3 upon dissolution in water gives Fe(OH)3 particles due to hydrolysis. Have you filtered/centrifuged iron cross linker solution to remove these particles?

3.           Line 102 – please, specify stirring speed and sizes of petri dish and stir bar

4.           Line 109 – it is great that the authors paid attention to the sterilization of beads, the procedure is often omitted in scientific papers. However, pasteurization does not kill spores. Doesn’t alginate withstand autoclaving? Did you try to autoclave solution of beads in water after preparation?

5.           Line 116-117 – please, check indentation

6.           Line 136 – Please, provide reference to literature on the method of film preparation.

7.           Line 207 – why did you choose non-parametric test (first of all for cell viability tests)? Is there any information in the literature on distribution of cell viability values when cells are grown in multiwell plates? Parametric tests are more powerful.

8.           Figure 2 – is it possible to quantify difference in cell growth between substrates (e.g. average fluorescence/number of cells per square or some other measure)? One image per substrate gives only limited qualitative picture.

9.           Figure 3 – it would be useful to support stability experiments with data on size change/swelling behavior of beads. In addition, original state of microbeads (prior to experiment) is not shown, therefore stability assessment is difficult

10.       Figure 5. A, C, E – please, use the same scale for Y and X axes for easier comparison.

11.       Figures 6, 7, and 9 – reporting standard deviation (SD is more informative then standard error of mean (SEM) for the assessment of between-sample variation or distribution of bead sizes, please provide SD values instead of SEM

12.       Figure 7. Difference between means is about 30%, however statistical analysis shows no difference. But I see that it is rather the result of very high variation in “no beads” samples. Is it possible to repeat the cell experiment with more replications (say 5) to obtain more accurate result? I mean that in the case of n=3 very small/very high value in one of three wells generates high variation and hinders statistical analysis. The same is for Figure 9, where SEM is between 45-85%, SD is therefore is about 5-95%? It looks like in one of the wells viability was almost zero, whereas in other two wells it was good enough. In other two bars, error is much smaller; therefore I think it could be the result of, say, contamination and not because of beads. I strongly recommend you to reproduce these experiments with more replications to obtain better evidence. It is important because it can affect conclusions of the paper.

13.       Figure 8B Ca-micro – number of cells decreased drastically from day 1 to day 7. Why? Quantitative analysis of cell proliferation would be useful and facilitate comparison (for all presented samples).

14.       Can the leaching of iron ions effect the cell growth?

Author Response

Reviewer 2 Report (New Reviewer)

Overall, a well-written manuscript on an interesting topic. There are a few minor things that need to be improved or explained. But there are also a few questions that may take a little more effort to answer. This is related to the lack of comparison of Ca alginate beads with Fe alginate beads in several images, although the manuscript is dedicated to showing that Fe beads are superior to Ca beads and instead a comparison between Fe beads and a plastic surface is given. Authors should explain why this was done or include the Ca bead data in their manuscript if these experiments were done or give an explanation as to why they were not done. In addition, although some graphs show that plastic is even better than Fe beads, the authors interpret these results as statistically insignificant. According to these claims, plastic is as good as Fe beads (if the obtained results are "statistically" interpreted in favor of plastic, because they can be even more accurate). Therefore, it must be stated that the obtained results show what they show and that additional, more detailed and extensive experiments must be done in order to clearly determine whether Fe beads are so superior, because some of the obtained results do not show statistical significance.

Comments are given in the PDF document.

Round 2

Reviewer 1 Report (New Reviewer)

The authors have addressed all the questions. 

Reviewer 2 Report (New Reviewer)

Now it may be published.

This manuscript is a resubmission of an earlier submission. The following is a list of the peer review reports and author responses from that submission.

Round 1

Reviewer 1 Report

This paper is very interesting in developing an electro spray to generate micro and macro alginate particles with Fe. This new way of generating particles and Fe coupled alginate samples could have potential in many areas. However, the key issue with this paper is that the authors claimed the Fe-alginate particles can improve MSC proliferation and growth, but the finding did not provide it. The cell viability is lower than plastic surface (Figure 9). Furthermore, there is no characteristics assessment on the MSCs that cultured to check they still maintain their properties (markers, functional test etc). This is the major drawbacks of this paper. It will also be good to provide more information on the film generation, as the experiment seems more like a coating surface not a fabrication of film. It is also unclear the day of staining in figure 2. Finally this discuss is weak, and there is lack of insights on the results. Why Fe alginate is better than Cal etc.  Perhaps the authors can think about using the Fe-alginate beads for other applications such as drug delivery.

Reviewer 2 Report

The authors described iron-crosslinked alginate beads for mesenchymal stem cell culture. Although some interesting data has been provided, I still have some questions.

1.       For the title, if the trials only applied MSCs, please specify it instead of using stem cells.

2.       For new novel technology applications, could the authors provide the comparison between the new technology beads with current commercial beads? And the reasons for using these beads.

3.       In cell culture part, could the authors add the passage number used in this experiment?

4.       GFP-MSCs were modified. Did the authors try human primary MSC culturing on these beads?

5.       In Figure 7, it looks like Cells alone condition is better than Cells with beads. Could the authors confirm there is no statistical significance in metabolic activity and viability? For cell growth comparison, the authors should use specific growth rate and population doubling time to quantify.

6.       After production of MSCs on these beads, the authors should provide the MSC potency, including colony formation, differentiation, and the surface markers to show if the MSCs still maintain their phenotype.

Reviewer 3 Report

The authors have investigated the cell attachment of an alginate preparation, that contained Fe3+ ions as the crosslinker.

Detailed opinion

Introduction

The Introduction is well written, however, there were no references that would justify the use of alginates as cell carriers neither as a pharmaceutical, nor as a medical device, which means that the regulatory related requirements are not clear.

Materials methods

The diameter distribution measurement needs to be described.

The meaning of ***: p<0.001, ***: p<0.0001 needs to be described.

The authors need to keep in mind that the preparation does not become sterile if they put it in sterile water, so the preparation is actually not sterile.

Line 108 and 108 needs editing.

Line 170 and 171 the sentence needs to be rephrased 

Results

Line 204 please use a more suitable word instead of "better"

Line 243 please rephrase "very important swelling"

Line 312 there are two *** symbols.